# Physical Chemical Investigation of Gamma-Irradiated Parchment for Preservation of Cultural Heritage

**DOI:** 10.3390/polym15041034

**Published:** 2023-02-19

**Authors:** Ion Bogdan Lungu, Lucretia Miu, Mihalis Cutrubinis, Ioana Stanculescu

**Affiliations:** 1Horia Hulubei National Institute for Physics and Nuclear Engineering (IFIN-HH), Multipurpose Irradiation Facility (IRASM Department), 30 Reactorului Str., 077125 Magurele, Romania; 2National Research & Development Institute for Textile and Leather (INCDT-ICPI), 93 Ion Minulescu Str., 030508 Bucharest, Romania; 3Department of Analytical Chemistry and Physical Chemistry, Faculy of Chemistry, University of Bucharest, 4-12 Regina Elisabeta Bld., 030018 Bucharest, Romania

**Keywords:** parchment, gamma, FTIR, mechanical tests, cultural heritage

## Abstract

The historical artefacts of parchment are prone to degradation if the storage conditions are improper due to the collagen structure having a limited stability under physical, chemical, and biological agent attacks. The parchment structure is difficult to characterize due to the variety of manufacturing traditions (eastern/western), intrinsic variability of skins (i.e., species, breeding variation, living conditions, effects of pathologies, etc.), biodeterioration, and aging, and the main concern in its analysis is its uniformity. The deterioration of parchment collagen produces a rather stiff or in some circumstances, a relaxed structure. Any intervention or treatment of unique, very precious cultural heritage artefacts must not negatively influence the properties of the component materials. Gamma irradiation is a relatively new technique of bioremediation. Data on the leather properties pre- and post-ionizing radiation bioremediation treatments are few in the literature. Fewer data are available on the historical leather and parchment physical chemical characteristics after ionizing gamma irradiation. This research had two main objectives: (i) the characterization of the parchment structure’s uniformity across the analyzed areas and its mechanical properties, i.e., tensile stress by mechanical tests and ATR-FTIR spectroscopy; and (ii) to establish parchment tolerance when exposed to ionizing gamma radiation as a pre-requisite for cultural heritage preservation irradiation treatment. It was found that the mechanical tests and ATR-FTIR spectroscopy may identify changes in the parchment’s irradiated structure and that the preservation of cultural heritage parchment artefacts may be performed at maximum 15 kGy gamma irradiation dose.

## 1. Introduction

Cultural heritage artefacts are prone to degradation due to physical, chemical, and biological factors. If physical and chemical degradation can be delayed by controlling the artefacts storage conditions, e.g., the temperature, relative humidity, and air purity, the biological attack, once installed, can only be stopped by a drastic intervention. Classical disinfection techniques based on using fumigation with various chemicals or solvents are ineffective against spores and eggs. New physical methods of conservation have been researched based on using lasers, plasma, or radiation treatments [1]. Among others, the ionizing radiation bioremediation treatment has the advantage of the certainty of the biocide effect, being a fast, mass, and safe treatment; no harmful chemicals and residues that may threaten the health of the museum personnel or general public are left in the materials [2]. Due to the novelty of the gamma irradiation treatment but also to the complexity and diversity of the constituents of cultural heritage items, the question that always remains to be answered is as follows: “Does the gamma radiation induces a supplementary degradation in the material?”. The subject is debated for numerous and diverse objects of wood [3], textile [4,5,6,7], paper [2,4,8,9,10,11,12,13], which are ornamented [14] or with polychromies [15,16,17,18]. It is also known that interventions for saving cultural heritage artefacts should be reversible. Nevertheless, the wood impregnation and consolidation using special resins and gamma radiation curing (Nucléart process), although irreversible, are used as cultural heritage artefacts salvation treatments [19]. In the literature, there are data on the behavior of the properties and mechanical parameters of leather under the effect of ionizing radiation, but there are fewer instances on historical examples, e.g., aged leather and parchment [20,21,22,23,24]. Parchment chemical structure is difficult to characterize due to its variability and aging conditions. The parchment is obtained from the skin of small animals such as sheep, goats, and calves through various treatments aimed at removing the hair by treatment with alkaline materials and drying in a tense state, and scraping and finishing by polishing and bleaching, in order to give the skin surface smoothness, suppleness, and opacity [25]. Although it is considered to be a primitive, coarse material, this is only in appearance, because the parchment has a sophisticated three-dimensional arrangement generated by its basic element, the collagen molecule, with a helical structure, which allows successive clockwise and counterclockwise windings and the formation of increasingly complex structures, such as collagen fibrils, collagen fibers, and three-dimensional collagen networks (e.g., derma, characterized by exceptional strength and durability). The millenary durability of the parchment is due, on one hand, to the simple chemical composition, but especially to this hierarchical structure of collagen, which is a complex structure that is difficult to evaluate as each hierarchical level requires a certain specific analysis technique, and a complete image of a parchment is obtained by correlating the results of all these techniques [26].

Skins display morphogenesis, which is formed from collagen molecules and diverse structures. The tanning of skins produces a crosslinked material while the parchment production generates a highly hygroscopic matrix due to the polar sites available for hydrogen bonding [27]. Microbiological infestation is common for parchment objects due to the presence of partially or completely gelatinized collagen fiber, and that is why safe storage implies dry conditions [27]. Chemical methods and freeze-drying were used to counter the mold growth on parchment. These methods may have adverse effects, producing changes in the material structure and in the case of the use of chemical fungicides, risks for one’s health and the environment may also appears. Gamma irradiation is researched as a new minimal invasive safe treatment technique, together with other physical techniques based on using plasma [28], lasers [29], e-beam, or X-rays [30,31,32]. The biocidal effect of gamma radiation is used for the sterilization of medical devices at industrial scale, which is effective for the whole volume of an artifact while plasma and e-beam techniques are used to treat surface attacks.

The main concern in the analysis of natural materials is their uniformity. One of the major problems when working with natural materials is the heterogeneity of the samples. In the case of skins, this is caused by the animal constitution and condition. Additionally, the parchment manufacturing process influences the structure of the final product. In order to exclude some of the mentioned influences, all the tested samples were collected from the middle section of one skin, close to the backbone of the animal, and the leather anatomical zone influence was proved by FTIR and mechanical tests. FTIR spectroscopy proved to be very useful in identifying the collagen structure and degree of conservation of various cultural heritage goods that were made of leather and parchment [33,34]. FTIR spectroscopy allows for the identification of changes at molecular and supramolecular levels by monitoring the band position and intensity. New bands in the FTIR spectrum, as well as the shift of band intensity, indicate the chemical or compositional modification of the samples, while the band position shifts indicate a conformation and/or crystallinity modification due to crosslinking, water adsorption/desorption, phase transitions, etc. [5,35]. The FTIR-ATR technique is commonly used in the study of collagenous structures allowing in-depth identification of alpha, beta, and random coil protein secondary structures [36,37]. Mechanical tests allow material characterization at a macroscale, and the combination of the FTIR analysis with mechanical test data may probe the correlation of molecular-scale modifications with the object’s macroproperties. This research had two main objectives: First, the characterization of the parchment’s structural uniformity across the analyzed areas and its mechanical properties, i.e., tensile stress. Secondly, to establish parchment tolerance when exposed to the ionizing gamma radiation as a pre-requisite for cultural heritage irradiation bioremediation treatment application. The motivation to treat the parchment at high gamma irradiation doses was simply to observe its tolerance to radiation. It was found that (i) mechanical tests and ATR-FTIR spectroscopy may identify changes in native parchment and gamma-irradiated structure; (ii) for cultural heritage parchment artefacts preservation the maximum allowed gamma treatment dose is 15 kGy. 

## 2. Materials and Methods

### 2.1. Parchment Preparation

The goat parchment was manufactured by INCDTP-ICPI, using traditional methods following the steps depicted in Figure 1. 

The processing operations of goat skins to obtain parchment are simple, only requiring washing, removal of hair and adhering residual flesh, and drying under tension. Washing is performed with Na_2_CO_3_ and house soap, and dehairing is performed with lime. Dehairing may take two days to one week based on the kind of skin and their curing state, environment temperature, and dehairing operation variables. In this case, the skin was dried for 3 days at normal temperature, between 22–28 °C, without ventilation in order to achieve a humidity of the material between 8–12%, depending on the relative humidity of the air. This humidity is specific to natural organic supports.

### 2.2. Gamma Irradiation of the Parchment

The samples were prepared for gamma irradiation and analysis at IFIN-HH, IRASM department, using a cutting tool according to ISO 37, type 4. The sampling was divided into 13 batches from the backbone to the belly area, as indicated in Figure 1; each batch contained 10 samples. The direction for collecting the samples was from head area to tail area. The direction of the samples was parallel to the backbone of the parchment skin. The samples were collected according to the parchment topography, i.e., starting from the backbone area downward to the belly area and from the head to the tail area. Figure 1 depicts the half goat skin parchment that was analyzed and the sample’s collection areas. Sample’s dimensions were measured using a Proma micrometer M026.

The samples were irradiated at IFIN-HH, IRASM department using a Cobalt-60 Gamma Cell 5000 research irradiator. There were 9 applied different irradiation doses: 3, 6, 10, 15, 20, 25, 35, 50, and 100 kGy, and 4 batches were kept as controls, at various distances from the backbone area to check parchment uniformity. The dose rate was approximately 6.51 kGy·h^−1^. The ethanol-chlorobenzene (ECB) dosimetry system was used to determine the applied dose and the dose uniformity. Between the irradiated batches were interleaved four non-irradiated batches (A, B, C, and D) in order to observe the uniformity and mechanical strength of the parchment. Non-irradiated batches were placed before 3 kGy and after 10, 25, and 100 kGy batches, as can be seen in Figure 1. Thus, their section intervals were obtained: interval no. 1 enclosing the batches: 0 kGy(A)–3 kGy–6 kGy–10 kGy–0 kGy(B); interval no. 2 enclosing the batches: 0 kGy(B)–15 kGy–20 kGy–25 kGy–0 kGy(C); and interval no. 3 comprising the batches: 0 kGy(C)–35 kGy–50 kGy–100 kGy–0 kGy(D).

### 2.3. Mechanical Tests

The mechanical tests were conducted on a Zwick Roell testing machine with a 5 kN cell force.

The ISO 3376:2011 standard “Physical and mechanical tests–Determination of tensile strength and percentage extension” was not used for these tests due to the large dimension of the dumbbell, which results in a small number of testing samples and a large variation of tensile stress within the sample due to the heterogenic characteristics of natural materials. Instead, ISO 37 standard: “Rubber, vulcanized or thermoplastic-Determination of tensile stress–strain properties” was used. The dumbbell has the following characteristics: overall length, 35 mm; test length, 10 ± 0.5 mm; and width, 2 ± 0.1 mm. The testing speed was 200 mm·min^−1^ with a pre-load of 1 N. The reason for using such a small dumbbell is that it provided a large number of samples, which decreased the measuring errors induced by the heterogeneity of the natural material. 

### 2.4. ATR-FTIR Spectroscopy Characterization

A second method of analysis was used, namely, FTIR by ATR technique, in order to observe the natural variability of the parchment structure and the possible changes at the molecular level after gamma irradiation, using a Bruker Vertex 70 machine, with a wavenumber range of 4000–400 cm^−1^, 4 cm^−1^ resolution, and diamond crystal ATR accessory. Ten measurements were conducted per sample and there were 256 scans per measurement. The results were mediated, thus a single spectrum resulted from the ten measurements. A total of eight control samples, four from the head area and four from the tail area, were collected from the extremes, head and tail, when furthering from the backbone to the belly. The samples were collected from the four interleaved non-irradiated batches corresponding to batch A, B, C, and D from head and tail areas to analyze anatomical position influence on the collagen parchment structure. In the case of the irradiated batches for each sample, there were also ten spectra measurements. 

## 3. Results and Discussion

### 3.1. Physical Chemical Properties of Unirradiated Parchment

Figure 2 presents the tensile stress mean values obtained for the four non-irradiated batch samples A, B, C, and D, and for exemplification, the original stress–strain curves for unirradiated samples (batch A). All the experimental data are available upon request from the authors.

The control batch A is placed closest to the backbone while the control batch D is closest to the belly area. The highest average value for the tensile stress is 39.4 N/mm^2^, which is closest to the backbone (batch A), while the lowest value is 28.9 N/mm^2^ (batch D), which is farthest from the backbone. The natural decrease in the tensile stress that characterizes the parchment’s skin topography is obvious, while the standard error of the mean is low, comprised between 3.05 and 2.16 N/mm^2^. The results indicate that the natural mechanical resistance decreases along with the growth of the distance from the backbone to the belly area, from batch A to batch D.

The ATR-FTIR spectra of the control batches from the head area sampling are shown in Figure 3. It can be observed that there is a decrease in the intensity of the FTIR spectrum when furthering from the backbone area (A) to the belly area (D), which is in agreement with the measured decreased tensile stress. High mechanical resistance of the sample provides a good contact with the ATR crystal and a stronger intensity spectrum, while the loose parchment structure of the belly area produces a less intense vibration band spectrum. In the parchment FTIR spectra, the following main collagen bands are observed: amide A (3300 cm^−1^), amide B (3086 cm^−1^), amide I (1632 cm^−1^), amide II (1539 cm^−1^), and amide III (1233 cm^−1^) [34]. The multiple maxima of amide I of the 1650 cm^−1^ band indicate the various collagen protein conformations: alpha, beta, or random structures. The alpha helix structures are depicted by the left maximum of the amide I band while the random structures are described by the right maximum. An increase in the random structures peak maxima intensity in the belly area was observed, which is in accord with the mechanical tests that showed a decrease in the mechanical strength, i.e., tensile stress (see inset of Figure 3).

The ATR spectra from the tail area are shown in Figure 4 for the unirradiated control batches. The same behavior with the head samples is observed for the tail area samples, e.g., the decrease in the intensity of organized helix structures is depicted by the left maximum, and increase in random coil and unorganized structures is depicted by the right amide I band maximum for samples farthest of the backbone. In conclusion, there is no difference in behavior related to the decrease in the mechanical properties of the parchment from the backbone to the belly area for the tail and head areas.

### 3.2. Physical Chemical Properties of Gamma-Irradiated Parchment

The results of the mechanical tests obtained for the irradiated samples were analyzed by looking at the three intervals defined by the non-irradiated batches (control group, A, B, C, and D) as follows (see Figure 5a–c): (a)Interval no. 1: 0 kGy (A)–3 kGy–6 kGy–10 kGy–0 kGy (B);(b)Interval no. 2: 0 kGy (B)–15 kGy–20 kGy–25 kGy–0 kGy (C);(c)Interval no. 3: 0 kGy (C)–35 kGy–50 kGy–100 kGy–0 kGy (D).

The control group batches (A, B, C, and D) were created in order to analyze the behavior of the tensile stress of the samples before and after the irradiation treatment, taking into account the natural decrease in the tensile stress induced by the topography (anatomical zone) of the skin parchment. 

The values for the irradiated samples at 3, 6, and 10 kGy, respectively, 36.2, 35.3, and 34.3 N/mm^2^ show a slight decrease from 3 to 6 and 10 kGy irradiation doses. Taking into consideration the calculated values of the standard error of the mean, respectively, 2.78, 3.13, and 2.79 N/mm^2^, one may conclude that the radiation has no damaging effect on the parchment because the observed decrease in the tensile stress is smaller than the natural decrease in the tensile stress in the direction from the backbone to belly area. Thus, the medium values for the irradiated samples are interposed between the non-irradiated batches, A and B samples, showing no influence of radiation treatment on the tensile stress. 

For the interval no. 2, the tensile stress of 31.2 N/mm^2^ for 15 kGy is above the non-irradiated batch B value of 31.7 N/mm^2^ when including the values of the standard error of the mean 1.66 N/mm^2^, respectively, 2.22 N/mm^2^. This result concludes no damaging effect of the 15 kGy radiation treatment. 

For the 20 and 25 kGy irradiation doses, the tensile stress values, respectively, 30.3 and 30 N/mm^2^ are below the non-irradiated second control batch B. Considering the values of the standard error of the mean 1.66, 1.83, and 1.81 N/mm^2^, we concluded that the gamma irradiation has a small damaging effect on the parchment, thus decreasing the tensile strength. 

For the interval no. 3, a damaging effect is observed at 100 kGy, and the value of 24.2 N/mm^2^ is lower than the natural decrease in tensile stress for the non-irradiated batch D, 28.9 N/mm^2^. Additionally, for the irradiated samples at 35 and 50 kGy, the tensile stress is below the non-irradiated control batch D (28.9 N/mm^2^), i.e., 28.2 and 27.9 N/mm^2^, respectively. 

For the irradiated samples, a slight decrease in the tensile stress can be considered a damaging effect, even if the standard error of the mean is taken into account, 1.75, respectively, 2.27 N/mm^2^. 

Figure 6 demonstrates the ATR FTIR spectra of the gamma-irradiated parchment samples.

The irradiated samples have similar FTIR spectra, as compared to the controls, and the collagen structure is conserved but some changes in the position, shape, and band intensity can be observed. For the lowest irradiation doses of each interval, the intensities of the right and left maximum of the amide I band are comparable, while at higher doses, some increase in the random coil structure’s peak component is observed, i.e., the right maximum component intensity of the amide I band increases with the irradiation dose.

The ATR technique results correlates well with the mechanical tests results, showing at molecular level the small structural differences for the irradiated and non-irradiated collagen parchment samples.

## 4. Conclusions

In this paper, the natural variability of the collagen parchment structure was demonstrated by mechanical testing and the ATR-FTIR spectra analysis. The change in the amide I band intensity and shape is correlated with small changes in the collagen protein conformations and correlates with the mechanical behavior. Additionally, the influence of the gamma irradiation treatment on the parchment’s collagen was identified from small changes in the ratio between the alpha helix and random coil secondary protein structures that form the left and right maxima of the amide I vibration band. The variation of the tensile stress was considered when establishing the maximum allowed gamma irradiation dose.

In conclusion, the gamma radiation treatment for parchment disinfection is safe for doses up to 15 kGy, as observed from small modifications of the ATR spectrum and mechanical tests. Taking into account the necessity of using maximum doses of 15 kGy, it can be stated that radiation treatment is a viable alternative/process for cultural heritage decontamination and preservation. Further investigation may concern the natural and/or artificial ageing influence on the structure and stability of gamma-irradiated parchment for an even safer approach for the preservation of unique cultural heritage artefacts.

## Data Availability

Data are available upon request from the authors.

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
