# Peer review of "Physical Chemical Investigation of Gamma-Irradiated Parchment for Preservation of Cultural Heritage"

_polymers, 2023, doi:10.3390/polym15041034_

Round 1

Reviewer 1 Report

This paper investigated the parchment structure uniformity across the analyzed areas and its mechanical properties, and parchment tolerance when exposed to ionizing gamma radiation as a pre-requisite for cultural heritage preservation irradiation treatment. This work may be important for the cultural heritage preservation, but I think it is not suitable for the publication in Polymers. Specific comments 1. The authors mentioned the sampling was divided into 13 batches, please check Figure 1 to be consistent. 2. The unit of tensile strength should be N/mm2, please revise all the same problem over the whole manuscript. Could the authors provide the original tensile stress-strain curves to replace Figures 2 and 5 for all the samples? 3. The figures are poor (i.e. Figure 3 and 4). 4. The analysis of Figure 6 should be revised. 5. The style of references should be consistent. 6. The grammar of the whole manuscript should be revised.

Reviewer 2 Report

The topic is interesting indeed, however there are several issues about this manuscript.

How many skins did the authors use? Did they ask for approval to an ethic committee??

Line 153 “The ISO 3376:2011 standard - „Physical and mechanical tests — Determination 153 of tensile strength and percentage extension” was not used…” What did the authors use instead?? What did they measure? Graphs axes have no units. Please use the same number at the axes in all the graphs.

Did the authors measure the thickness of the samples? Such parameter could be important for mechanical resistance, especially for parchment topography.

How long were the skins dried? Was it the same for all the skins?? That parameter could be important for gamma irradiation.

Redaction and grammar have to be checked out; there are several mistakes throughout the manuscript.

Round 2

Reviewer 1 Report

The authors have revised the manuscript as required.